# Searching for Efficient Linear Layers over a Continuous Space of Structured Matrices

**Andres Potapczynski**\*
New York University

**Shikai Qiu**\*
New York University

**Marc Finzi**
Carnegie Mellon University

**Christopher Ferri**
Capital One

**Zixi Chen**
New York University

**Micah Goldblum**
Columbia University

**C. Bayan Bruss**
Capital One

**Christopher De Sa**
Cornell University

**Andrew Gordon Wilson**
New York University

## Abstract

Dense linear layers are the dominant computational bottleneck in large neural networks, presenting a critical need for more efficient alternatives. Previous efforts focused on a small number of hand-crafted structured matrices and neglected to investigate whether these structures can surpass dense layers in terms of compute-optimal scaling laws when both the model size and training examples are optimally allocated. In this work, we present a unifying framework that enables searching among all linear operators expressible via an Einstein summation. This framework encompasses many previously proposed structures, such as low-rank, Kronecker, Tensor-Train, Block Tensor-Train (BTT), and Monarch, along with many novel structures. To analyze the framework, we develop a taxonomy of all such operators based on their computational and algebraic properties and show that differences in the compute-optimal scaling laws are mostly governed by a small number of variables that we introduce. Namely, a small $\omega$ (which measures parameter sharing) and large $\psi$ (which measures the rank) reliably led to better scaling laws. Guided by the insight that full-rank structures that maximize parameters per unit of compute perform the best, we propose BTT-MoE, a novel Mixture-of-Experts (MoE) architecture obtained by sparsifying computation in the BTT structure. In contrast to the standard sparse MoE for each entire feed-forward network, BTT-MoE learns an MoE in every single linear layer of the model, including the projection matrices in the attention blocks. We find BTT-MoE provides a substantial compute-efficiency gain over dense layers and standard MoE.

## 1 Introduction

Neural networks primarily consist of interleaved linear layers and simple non-linearities. In large foundation models such as GPT-3 [4], these linear layers consume the vast majority of the parameters and computation [13], and are commonly represented by dense matrices. Substituting these dense matrices with structured matrices with fast matrix-vector multiplies (MVMs) has the potential to significantly improve the computational efficiency of these models.

Recent work by Dao et al. [6], Fu et al. [8], and Qiu et al. [19] demonstrated that incorporating certain structured matrices into neural network architectures, including transformers, can improve

---

\*Equal contribution. Correspondence to `ap6604@nyu.edu`, `sq2129@nyu.edu`, `andrewgw@cims.nyu.edu`.

38th Conference on Neural Information Processing Systems (NeurIPS 2024).

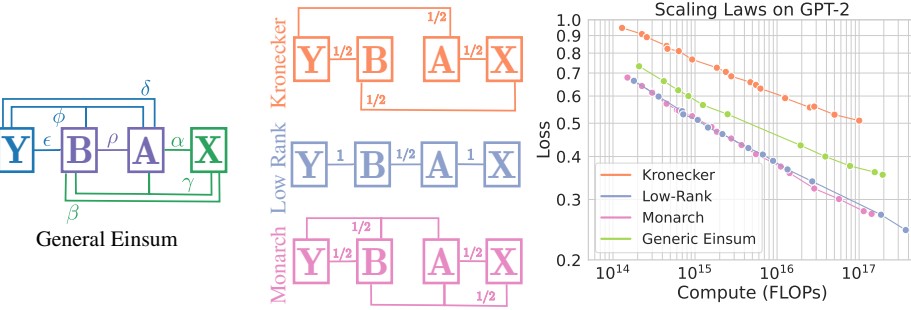

Figure 1: **We use Einsums to parameterize a wide range of structured matrices and search for the most efficient structure for compute-optimal training. Left:** A diagrammatic representation of a general two-factor Einsum. We parameterize the space of Einsums through a real-valued vector $\boldsymbol{\theta} = (\theta_\alpha, \theta_\beta, \theta_\gamma, \theta_\delta, \theta_\epsilon, \theta_\phi, \theta_\rho) \in [0, 1]^7$. This space captures many well-known structures through specific values of $\boldsymbol{\theta}$. **Middle:** Example of well-known structures with their $\boldsymbol{\theta}$ values. Any omitted line implies the value of the entry in the vector is 0. **Right:** Compute-optimal scaling laws of example structures for GPT-2 on OpenWebText when substituting its dense layers (see details in Section 4).

performance over dense models of the same size trained for equal number of epochs on problems such as ImageNet classification. However, these success cases do not reflect the current paradigm of large-scale training, where the models 1) are typically not trained for multiple epochs, making the expressiveness of dense matrices particularly appealing since the generalization gap vanishes, and 2) are heavily bottlenecked by compute cost, making it infeasible to train the models until convergence [13, 11], unlike in image classification. These attributes of large-scale training make the compute-optimal scaling rather than scaling in model size alone more relevant.

In this work, we investigate how different structures perform in a compute-optimal setting, which characterizes performance as a function of training compute when allocated optimally between using larger models versus training on more data [13]. In language modeling and many other tasks, the compute-optimal scaling law has been shown to take the form $\mathcal{L} = \mathcal{L}_\infty + bC^{-a}$ as a function of training compute $C$, where $\mathcal{L}_\infty$ is the minimal achievable loss [13, 11, 10]. Quantifying the compute-optimal scaling laws of various structures is essential for understanding their practical value for training large-scale neural networks.

In addition to investigating the scaling laws of existing structures, we expand the set of matrix structures beyond what has been previously considered. We do so by introducing a continuous parameterization of the space of all possible structures whose matrix-vector-multiplication (MVM) can be expressed as an Einstein summation (Einsum).[2] This space contains many known structures such as low-rank, Tensor-Train [17], Kronecker product [21, 25, 9], Monarch [6] and Block Tensor-Train [19], but also includes many novel hardware-efficient structures. Indeed, all structures in this space are hardware-efficient in the sense that they are computed through a series of batch matrix multiplication primitives, which we implement through the `Linear Operator` abstractions available in `CoLA` [18]. Moreover, this space lends itself to an intuitive exploration as we can analyze how different parameters of the Einsum affect a structure's performance and scaling laws. We make our code available here.

We summarize our main contributions as follows:

- We introduce a continuous parameterization of the space of structured matrices whose matrix-vector-multiplication can be implemented via an Einstein summation (Einsum). This parameterization allows us to search a wide range of hardware-efficient structured linear layers for neural network architectures beyond a handful of well-known cases identified in prior work [6, 8, 19].

- We develop a taxonomy of the space of Einsums based on its computational and algebraic properties. We identify three key scalar quantities that characterize this space $(\omega, \psi, \nu)$. (1) $\omega \geq 0$, which reflects the extent of parameter sharing in a matrix. (2) $\psi \in [0, 1]$, which characterizes to the rank of the structure ($\psi = 1$ meaning full-rank). (3) $\nu \in [0, 1)$, which

---

[2]Technically, we consider everything that can be expressed via `torch.einsum`, which is slightly more general than the Einstein summation, which allows an index to appear at most twice.

relates to compute per dimension in an MVM, where the upper-bound $\nu = 1$ is achieved by dense matrices. Intuitively, $\nu$ measures how much a structure resembles dense.

- We investigate the scaling laws of different Einsums on language modeling, autoregressive image generation, and a synthetic regression task. We find that the best-performing structures are the ones that do not share parameters ($\omega = 0$), are full-rank ($\psi = 1$), while $\nu$ can be varied with often negligible impact to their scaling laws. In contrast to previous findings, we demonstrate that structures shown in prior work [6, 8, 19] to outperform dense matrices in non-compute-optimal settings can yield similar but not significantly better compute-optimal scaling laws on these tasks.

- Building on prior work in structure-aware learning rates [19], we show how to properly initialize generic Einsum layers and transfer learning rates from the original dense layers and across model sizes, leveraging insights from $\mu$P [29, 26] and manifold optimization [3].

- Based on the observed relation between the taxonomy variables and the scaling laws, we propose a new structured Mixture of Experts (MoE) architecture implementing a sparse mixture of multiple structure matrices. This block tensor-train (BTT) MoE provides a sparse MoE in every single layer of each feedforward network (FFN) and attention project matrices, compared to standard MoE which operates over entire FFNs. We show BTT-MoE is significantly more compute-efficient than dense matrices and standard MoE for training GPT-2 language models.

## 2  Parameterizing the Space of Einsums

We now present a unifying framework that parameterizes all linear operators $\mathbf{W} \in \mathbb{R}^{d_\text{out} \times d_\text{in}}$ whose matrix-vector-multiply $\mathbf{y} = \mathbf{W}\mathbf{x}$ can be expressed as an Einsum over the tensors $\mathbf{x}, \mathbf{A}, \mathbf{B}, \ldots$, where $\mathbf{A}, \mathbf{B}, \ldots$ defines the operator $\mathbf{W}$ and contains all its learnable parameters. To simplify the presentation, throughout this paper we assume $\mathbf{W}$ is defined using only two factors $\mathbf{A}, \mathbf{B}$, but we show generalization to more than two factors is straightforward in Appendix E.

We consider the following general expression of such an Einsum

$$Y_{\delta\epsilon\phi} = \sum_{\alpha\beta\gamma\rho} B_{\beta\gamma\epsilon\phi\rho} A_{\alpha\gamma\delta\phi\rho} X_{\alpha\beta\gamma}, \tag{1}$$

where the vectors $\mathbf{x}$ and $\mathbf{y}$ are written as tensors with multiple indexes to allow them to interact differently with each other, and the factors $\mathbf{A}, \mathbf{B}$. Each index $x \in \{\alpha, \beta, \gamma, \delta, \epsilon, \phi, \rho\}$ ranges from 1 to $d_x$. Given this general expression, we obtain different structures via different factorizations of $d_\text{in}$ into $d_\alpha d_\beta d_\gamma$, and $d_\text{out}$ into $d_\delta d_\epsilon d_\phi$, and separately a choice of $d_\rho$. For example, for a low-rank matrix, we have $d_\alpha = d_\text{in}, d_\beta = d_\gamma = 1$, $d_\epsilon = d_\text{out}, d_\delta = d_\phi = 1$ and $d_\rho = r$. For a Kronecker product, we have $d_\alpha = d_\beta = \sqrt{d_\text{in}}$, $d_\delta = d_\epsilon = \sqrt{d_\text{out}}$ and $d_\gamma = d_\phi = d_\rho = 1$. We provide an extended list of examples in Appendix A.

The general expression and the above two examples can be more conveniently and intuitively represented as a diagram shown in Figure 1 (left), where each index $\alpha, \beta, \gamma, \ldots$ corresponds to a (hyper)edge among the input, output, and weight factors: $\alpha \leftrightarrow \{\mathbf{X}, \mathbf{A}\}$, $\beta \leftrightarrow \{\mathbf{X}, \mathbf{B}\}$, $\gamma \leftrightarrow \{\mathbf{X}, \mathbf{A}, \mathbf{B}\}$, $\delta \leftrightarrow \{\mathbf{Y}, \mathbf{A}\}$, $\epsilon \leftrightarrow \{\mathbf{Y}, \mathbf{B}\}$, $\phi \leftrightarrow \{\mathbf{Y}, \mathbf{A}, \mathbf{B}\}$ and $\rho \leftrightarrow \{\mathbf{A}, \mathbf{B}\}$. This set of edges can be written succinctly as $\mathcal{E} = \{S \subseteq \{\mathbf{X}, \mathbf{A}, \mathbf{B}, \mathbf{Y}\} : |S| \geq 2 \text{ and } \{\mathbf{X}, \mathbf{Y}\} \not\subseteq S\}$. We exclude subsets that contain $\mathbf{X}$ and $\mathbf{Y}$ simultaneously, as adding them simply produces an already included structure but repeated multiple times along one of the input and output axes. The structure of a particular Einsum is fully specified by the the vector $(d_{\mathbf{X}\mathbf{A}}, d_{\mathbf{X}\mathbf{B}}, d_{\mathbf{X}\mathbf{A}\mathbf{B}}, d_{\mathbf{Y}\mathbf{A}}, d_{\mathbf{Y}\mathbf{B}}, d_{\mathbf{Y}\mathbf{A}\mathbf{B}}, d_{\mathbf{A}\mathbf{B}})$, which specifies the range of the indices $\alpha, \beta, \gamma, \delta, \epsilon, \phi, \rho$. When the range of an index is of size 1, the corresponding edge effectively disappears from the diagram and the expression simplfies.

As we will build models of varying sizes, it is more natural to think about how these entries scale with $d_\text{in}$ and $d_\text{out}$. We therefore assign a real-valued vector $\boldsymbol{\theta} \in [0, 1]^7$ to each structure indicating that $d_i = d_\text{in}^{\theta_i}$ for $i \in \{\mathbf{X}\mathbf{A}, \mathbf{X}\mathbf{B}, \mathbf{X}\mathbf{A}\mathbf{B}\}$, $d_j = d_\text{out}^{\theta_j}$ for $j \in \{\mathbf{Y}\mathbf{A}, \mathbf{Y}\mathbf{B}, \mathbf{Y}\mathbf{A}\mathbf{B}\}$, and $d_{\mathbf{A}\mathbf{B}} = \min(d_\text{in}, d_\text{out})^{\theta_{\mathbf{A}\mathbf{B}}}$, with the restriction that $\theta_{\mathbf{X}\mathbf{A}} + \theta_{\mathbf{X}\mathbf{B}} + \theta_{\mathbf{X}\mathbf{A}\mathbf{B}} = \theta_{\mathbf{Y}\mathbf{A}} + \theta_{\mathbf{Y}\mathbf{B}} + \theta_{\mathbf{Y}\mathbf{A}\mathbf{B}} = 1$. For example, a low-rank matrix whose rank scales as the dimension it operates on to the $1/2$-th power is represented as $\boldsymbol{\theta} = (1, 0, 0, 0, 1, 0, 1/2)$, and a Kronecker product of two factors of equal sizes is represented as $\boldsymbol{\theta} = (1/2, 1/2, 0, 1/2, 1/2, 0, 0)$. We round all $d_i$ to its nearest integer when

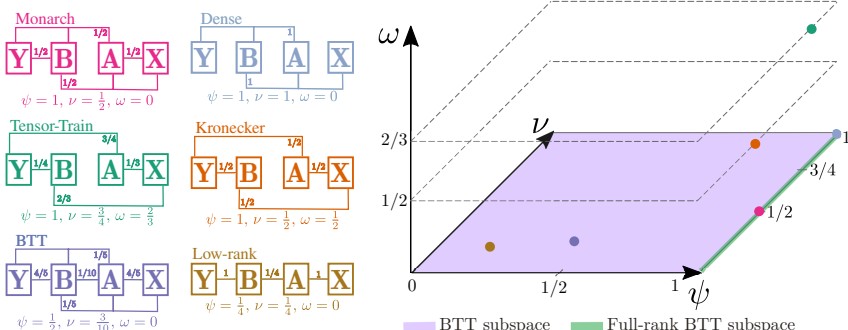

Figure 2: **Illustrating the Einsum taxonomy.** The 3D graph represents relevant quantities of the Einsum structure such as the amount of parameter sharing $\omega$ (x-axis), its rank $\psi$ (y-axis), and its compute intensity $\nu$ (z-axis). The structures on the left of the figure appear as dots on the graph based on their coordinates $\boldsymbol{\theta}$. We highlight two key subspaces. (a) The BTT subspace, characterized by no parameter sharing $\omega = 0$, learning the maximum number of parameters per FLOP. (b) The full-rank BTT subspace where $\omega = 0$ and $\psi = 1$. In Section 4 we show that the full-rank BTT subspace contains the most performant structures across multiple tasks.

instantiating the Einsum. Note this rounding only quantizes the the values of $d_i$, but leaves the space of meaningfully different $\boldsymbol{\theta}$s continuous as we consider arbitrarily large matrices.

## 3 A Taxonomy of the Space of Einsum Linear Structures

The space of all Einsums is a high-dimensional space containing a wide range of possible structures. *A priori*, it is difficult to reason about the properties of a particular point in this space given its coordinates $\boldsymbol{\theta} \in [0,1]^7$. In this section, we develop a taxonomy of the space of Einsums based on their computational and algebraic properties. We introduce three key scalar quantities that characterize this space: (1) $\psi \in [0,1]$, which is related to the rank of the structure, (2) $\nu \in [0,1]$, which is related to the compute intensity of the structure, i.e. FLOPs per MVM divided by the dimension, and (3) $\omega \geq 0$, which is related to the number of learnable parameters divided by the FLOPs per MVM. Each of these three quantities can be expressed in terms of the entries of $\boldsymbol{\theta}$, which we derive in Appendix B. To simplify the presentation, we assume $d_{\text{in}} = d_{\text{out}} = d$ in the rest of the section. Without loss of generality, we assume $\min(\theta_{\mathbf{XA}}, \theta_{\mathbf{YB}}) \geq \min(\theta_{\mathbf{YA}}, \theta_{\mathbf{BX}})$, so that it is more efficient to first multiply by $\mathbf{A}$ instead of $\mathbf{B}$.

**Rank Exponent, $\psi$.** For a given $\boldsymbol{\theta}$, we have that $\text{rank}(\mathbf{W}) = \Theta\left(d^\psi\right)$ where $\psi = \min(1, 2 + \theta_{\mathbf{AB}} - \theta_{\mathbf{XA}} - \theta_{\mathbf{YB}})$. Thus, $\psi = 1$ implies full-rank and decreasing values of $\psi$ imply lower ranks until the limit of 0. In other words, the rank decreases when an Einsum increasingly allocates part of the input and output to factors that are *only* connected the input or output. The limit being a low-rank structure as seen in Figure 2 where $\theta_{\mathbf{XA}} = 1 = \theta_{\mathbf{YB}}$ and which creates a bottleneck on $\{\mathbf{A}, \mathbf{B}\}$. The opposite trend is exhibited by dense, which allocates the full dimension of the input and output to both factors as seen in Figure 2. Nonetheless, it is still possible to achieve a full-rank when allocating dimension to only single factors as long as those values do not constitute most of the allocation, meaning $0 < \theta_{\mathbf{XA}} \leq 1/2$ and $0 < \theta_{\mathbf{YB}} \leq 1/2$, as seen in Monarch, Tensor-Train, and Kronecker

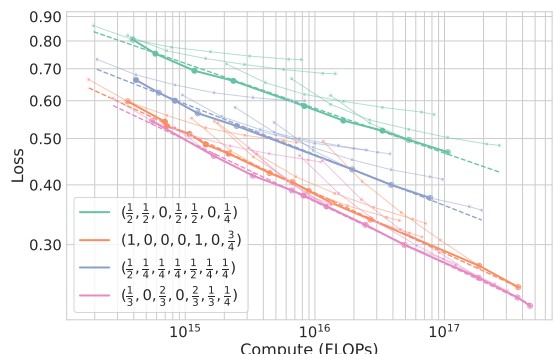

Figure 3: **Compute-optimal frontier (highlighted points) of various Einsums follows power law scaling.** As a result, Einsums can be scaled to reach arbitrarily low reducible loss, each with a different rate that can be estimated from small-scale experiments.

but not for the particular Block Tensor-Train (BTT) structure [19] in Figure 2. In Section 4 we show that structures with $\psi = 1$ perform best when training neural networks.

**Compute Intensity Exponent, $\nu$.** Let $F$ denote the FLOPs required to perform a MVM and define the compute intensity as $F/d = \Theta(d^\nu)$. The upper bound is achieved by dense, which requires quadratic compute for an MVM and thus $\nu = 1$. In general, we have $\nu = 1 + \theta_{\mathbf{AB}} - \min(\theta_{\mathbf{XA}}, \theta_{\mathbf{YB}})$. Thus, in order to achieve lower compute intensity than dense, a structure has to allocate dimensionality to factors that *only* connect to the input and output. As seen in Figure 2, the BTT example is able to achieve the lowest compute intensity by allocating a substantial part of the input dimension to the first factor and a substantial part of the output dimension to the second factor. $\psi$ and $\nu$ are not completely independent, e.g. $\psi = \nu$ for low-rank matrices $W_{ij} = \sum_{k=1}^r B_{ik} A_{kj}$, though exceptions exist such as for the Kronecker product where $\nu$ can be arbitrarily low while maintaining $\psi = 1$. In Section 4, we show there exists a wide range of structures with varying $\nu$ that perform as well as dense matrices.

**Parameters-Sharing Exponent, $\omega$.** Let $N$ denote the number of parameters in the structure then $N/F = \Theta(d^\omega)$. Clearly, $\omega = 1$ for dense matrices where each parameter is used exactly once in an MVM. In general, we can show that $\omega = \min(\theta_{\mathbf{XA}} + \theta_{\mathbf{YA}}, \theta_{\mathbf{XB}} + \theta_{\mathbf{YB}}) - \min(\theta_{\mathbf{XA}}, \theta_{\mathbf{YB}})$. In Section 4 we find that structures that share parameters, that is $\omega > 0$, have worse scaling laws that structures that do not ($\omega = 0$). To achieve $\omega = 0$, we have to avoid introducing edges that *skip* some factors, that is $\theta_{\mathbf{XB}} = \theta_{\mathbf{YA}} = 0$. Structures that skip factors in Figure 2 are Tensor-Train and Kronecker where there exists an edge that connects $\mathbf{X}$ to $\mathbf{B}$ while skipping $\mathbf{A}$. In contrast, in Monarch and BTT, the edge connecting $\mathbf{X}$ with $\mathbf{B}$ also touches $\mathbf{A}$.

## 4 Scaling Laws of Einsums

While prior works have shown that certain structured matrices such as Monarch and BTT have better scaling laws than dense matrices as a function of model size on datasets such as CIFAR-10, CIFAR-100, and ImageNet [6, 19], their experimental setups do not reflect today's large-scale training, where the models 1) typically do not train for multiple epochs on the training set, and 2) are heavily compute-bottlenecked such that we care primarily about performance as a function of training compute rather than model size (omitting the cost of training). These attributes of large-scale training make the compute-optimal scaling rather than scaling in model size alone more relevant.

In this section, we investigate the compute-optimal scaling laws of a wide range of Einsums — how their performance scales as a function of training compute. We will show that we can understand the systematic differences in the scaling laws of various Einsums by leveraging the taxonomy we have developed. While we do not find a structure that achieves noticeably better scaling laws compared to dense matrices, we identify the set of common properties shared across a wide range of structures that match the performance of dense matrices, based on which we will propose a significantly more efficient alternative to dense layers in Section 5.

### 4.1 Main Experimental Setup

We train GPT-2 [20] language models on the OpenWebText dataset. To make our measurement of the scaling laws more robust and our experiments more affordable, we reduce the vocabulary of the original GPT-2 to 96 commonly used alphanumeric symbols. Using a small vocabulary limits the compute and parameters consumed by the language modeling head, which would otherwise obscure the scaling laws measured at small scales [13]. We train models of varying sizes from 120k to 76M parameters, with model dimension $d \in [256, 4096]$ and number of transformer blocks $L \in \{3, 6\}$. Each model is trained for 100k steps with a batch size of 65536 tokens and a sequence length of 128. All linear layers except the language modeling head are replaced with Einsums. We use the Adam optimizer with a base learning rate of 0.003 for a $L = 3, d = 256$ dense model, and scale it using $\mu$P [27] and structure-aware learning rates [19] to larger models and models using Einsums in place of dense layers. We discuss learning rate scaling in detail in Section 6, showing it is crucial for the performance of Einsums. We use weight normalization to stabilize the training of Einsums following Qiu et al. [19].

In Appendix Appendix D, we show our main conclusions derived from this simplified setup translate to the more standard GPT-2 evaluation with a longer sequence length of 512 and its original vocabulary of 50,257 tokens.

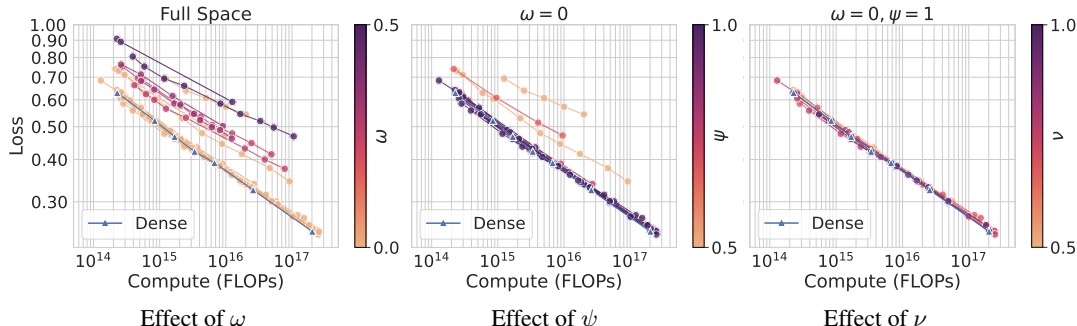

Figure 4: **The taxonomy parameters** $(\omega, \psi)$ **explain differences in the scaling laws.** (**Left**): parameter sharing ($\omega > 0$) leads to worse scaling. (**Middle**): among structures without parameter sharing ($\omega = 0$), full-rank structures ($\psi = 1$) scale better than low-rank structures ($\psi < 1$). (**Right**): in the ($\omega = 0, \psi = 1$) subspace, various structures have nearly indistinguishable scaling laws compared to dense matrices, suggesting that not implementing parameter sharing and being full-rank are the necessary and sufficient conditions for a compute-efficient linear layer for GPT-2.

## 4.2 Analyzing the Compute-Optimal Scaling Laws

**Einsum Performance Obeys Power Law Scaling.** When replacing the standard dense layers with Einsums, we find the resulting model's loss continues to follow the usual $\mathcal{L} = \mathcal{L}_\infty + bC^{-a}$ compute-optimal scaling laws except with possibly different constants $a, b$. In Figure 3, we visualize the compute-optimal scaling laws of various Einsums on our language modeling task, including those corresponding to previously proposed structures such as TT [17], Low-rank and BTT [19], as well as a generic Einsum with all entries of $\boldsymbol{\theta}$ strictly positive. We report the reducible loss with an estimated $\mathcal{L}_\infty = 0.75$ subtracted. This finding suggests that all Einsums can be scaled to reach arbitrarily low reducible loss, each with a different rate that can be estimated from small-scale experiments.

**Parameter Sharing Leads to Worse Scaling.** As discussed in Section 3, the vast majority of Einsums implement some kind of parameter sharing, where the number of parameters $N$ in the Einsum relates to its MVM FLOPs $F$ via $N/F = \Theta(d^{-\omega})$, for some $\omega > 0$. In Figure 4 (left), we show the scaling laws of a wide range of Einsums (only including points on the compute-optimal frontier) colored by $\omega$. We find larger values of $\omega$ lead to significantly worse scaling laws. To search for compute-efficient structures, we should therefore focus on the subspace with $\omega = 0$.

**Full-Rank Performs Best.** Within the $\omega = 0$ subspace, we find that $\psi$ becomes the next most important parameter. Recall $\psi \in [0, 1]$ is defined such that the rank of the Einsum scales as $\Theta(d^\psi)$. Einsums with $\psi < 1$ introduce information bottlenecks in the model by preventing the linear layers from accessing information from all the feature dimensions. The smaller $\psi$ is, the more severe this effect. In Figure 4 (middle), we show that small values of $\psi$ indeed lead to worse scaling laws. This observation further narrows down our search to the subspace with $\omega = 0$ and $\psi = 1$, i.e. the space of full-rank BTT matrices.

**Any Full-Rank BTT Scales Similarly as Dense.** The $\omega = 0$ and $\psi = 1$ subspace contains the Monarch matrices and its generalization BTT matrices[3]. From a computational perspective, a primary distinguishing factor among these structures is how close they resemble a dense matrix, which we characterize by their compute intensity $\nu \in [0, 1)$ defined so that $F/d = \Theta(d^\nu)$. $\nu$ is large whenever their exists large values (close to 1) in the remaining allowed entries ($\theta_{\mathbf{AX}}, \theta_{\mathbf{ABX}}, \theta_{\mathbf{YB}}, \theta_{\mathbf{YAB}}, \theta_{\mathbf{AB}}$). In Figure 4 (right), we show that, somewhat surprisingly, $\nu$ has minimal effect on the scaling laws of these structures. Structures with different $\nu$ have almost indistinguishable scaling laws compared to each other and dense matrices, which has $\nu = 1$. This result shows that while dense matrices perform well compared to the vast majority of possible Einsums, their good performance does not arise from being dense, but rather from not sharing parameters and being full-rank.

**Reconciling with Results from Prior Work.** Our findings do not mean that structured matrices cannot outperform dense in other settings. Rather, they highlight that the relative performance between structures depends on what resource is controlled, as prior work has shown that low rank, Tensor-

---

[3]In the 2-factor case, this space corresponds to all Block Low-Rank [1] matrices that are full-rank.

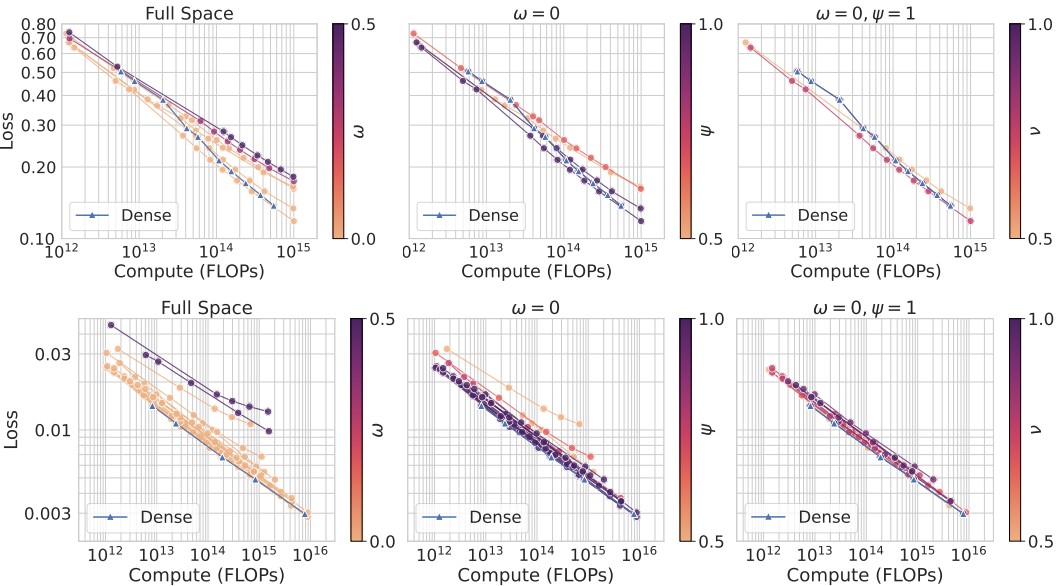

Figure 5: **Our findings about the effect of $(\omega, \psi, \nu)$ on the scaling laws generalize to other settings.** (**Top row**) Transformers trained with cross-entropy for autoregressive pixel generation on CIFAR-5M. (**Bottom row**) MLP trained with mean-squared-error loss on synthetic data generated by a large and randomly initialized MLP.

Train, Monarch, and BTT can significantly outperform dense in other settings such as controlling for memory, model size, or inference compute [5, 6, 30, 17, 19, 15], rather than training compute. For example, when training dataset size instead of training compute is the primary bottleneck, such as on conventional vision datasets like CIFAR-10 and ImageNet, structured matrices have shown considerable advantage over dense as a function of model size and inference compute [6, 19, 14]. In those settings, Qiu et al. [19] observe the benefits of structure likely arise through enabling computationally efficient wider layers.

### 4.3 Our Findings Generalize to Other Settings

We now test if our findings derived from the GPT-2 experiments can generalize to other settings. We evaluate on the following two additional tasks where there is sufficient data to measure the compute-optimal scaling laws without repeating training data. We provide additional experiment details in Appendix D.

**Autoregressive Pixel Modeling on CIFAR-5M.** We train transformers to autoregressively predict the RGB pixel values of images in the CIFAR-5M dataset [16], downsampled to $8 \times 8 \times 3$ resolution. Figure 5 (top row) shows qualitatively the same results as our GPT-2 experiments, where $\omega$ and $\psi$ have the most significant impact on the scaling laws, while varying $\nu$ yield only slight variations. In this particular case, having $\nu = 0.75$ (BTT with BTT-rank scaling as $d^{1/4}$) is better than having $\nu = 0.5$ (Monarch matrices). One notable trend in this setup is that most Einsums, regardless of $\omega$ or $\psi$, outperform dense at small scales. We hypothesize this improved performance is due to Einsums having larger embedding dimensions than dense layers for a fixed parameter budget and can thus preserve more information about the input pixels at smaller model sizes.

**MLP Regression on Synthetic Data.** We train MLPs on a synthetic regression dataset where the target is a scalar-valued function defined by a large randomly initialized MLP, similar to the student-teacher setting in [2]. In Figure 5 (bottom row), we observe qualitatively the same results as in the GPT-2 experiments.

Together, these additional results suggest there is some degree of universality associated with our findings on the effect of $\omega$, $\psi$, and $\nu$ on the compute-optimal scaling laws of neural networks that use Einsums in place of dense matrices.

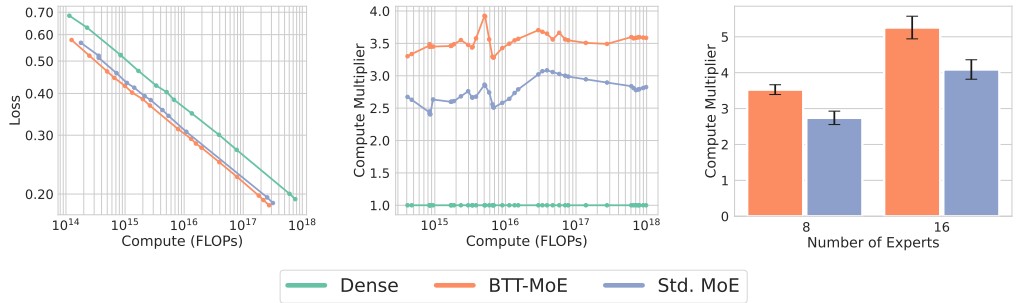

Figure 6: **BTT Mixture-of-Experts has significantly better compute-optimal scaling laws than dense GPT-2 and its standard MoE variant.** (**Left**): Compute-optimal frontier with 8. (**Middle**): 8 experts compute multiplier of BTT-MoE and standard MoE relative to dense as a function of FLOPs required by the dense model to achieve the same loss. (**Right**): Increasing the number of experts improves computational savings. Mean and standard deviation of the compute multiplier over all compute observations for 8 and 16 experts.

## 5    Structured Mixture of Experts

In Section 4, we identified that Einsums with $\omega = 0$ and $\psi = 1$ perform the best, and $\omega > 0$ or $\psi < 1$ lead to worse-than-dense performance. The most impactful parameter on the scaling laws is $\omega$, which measures how many parameters an Einsum has compared to the FLOPs for an MVM. Einsums that learn one parameter per FLOP perform significantly better than those that learn less than one parameter per FLOP. Therefore, a natural question arises: can we design structures that learn more than one parameter per FLOP, which we might expect will have even better scaling laws? Doing so requires that not all parameters are used in an MVM, which necessitates a sparse Mixture-of-Experts (MoE) like architecture [22, 7, 12]. Furthermore, we would like the structure to be full-rank, i.e. $\psi = 1$. In the following section, we introduce such a structure and demonstrate significant improvement over dense layers and the standard MoE architecture for training GPT-2.

### 5.1    More Parameters than FLOPs via Mixture of Experts

One natural candidate for constructing such a layer via an Einsum is to turn a BTT with BTT-rank $E$, which involves a sum over the rank index $\rho = 1, \ldots, E$ :

$$Y_{\epsilon\phi} = \sum_{\alpha\gamma\rho} B_{\gamma\epsilon\phi\rho} A_{\alpha\gamma\phi\rho} X_{\alpha\gamma} \tag{2}$$

into a $k$-sparse sum:

$$Y_{\epsilon\phi} = \sum_{\rho} g_{\rho} \underbrace{\sum_{\alpha\gamma} B_{\gamma\epsilon\phi\rho} A_{\alpha\gamma\phi\rho} X_{\alpha\gamma}}_{\text{output of } \rho\text{-th expert}}, \tag{3}$$

where $\mathbf{g} \in \mathbb{R}^E$ is a $k$-sparse vector so that only $k$ out of $E$ terms need to be computed. We interpret $k$ as the number of active experts and $E$ as the total number of experts. We compute $\mathbf{g}$ via a softmax over the top-$k$ entries of the logits $\mathbf{e}$ produced by a (dense) linear gating function $\mathbf{e} = \text{Linear}(X) \in \mathbb{R}^E$. There is no need to make this gating function structured because its cost is negligible. We choose $\theta_{\mathbf{AX}} = \theta_{\mathbf{ABX}} = \theta_{\mathbf{YB}} = \theta_{\mathbf{YAB}} = 1/2$ so that each expert is full-rank. We follow the common practice of using $k = 2$. The resulting BTT-MoE layer is a BTT with BTT-rank 2 (sum of two Monarch matrices) with input-dependent parameters. It is similarly straightforward to construct structured MoE from other structures by sparsifying the sum over $\rho$ with a gate. We use the load-balancing loss to encourage equal utilization of all experts [22, 7, 23].

In contrast to the standard MoE architecture used in transformer language models, which uses a sparse MoE for each entire feed-forward network (FFN) [22, 7, 12]:

$$\mathbf{Y} = \sum_{i=1}^{E} g_i \underbrace{\mathbf{W}_i^{\downarrow}\text{ReLU}(\mathbf{W}_i^{\uparrow}\mathbf{X})}_{\text{output of } i\text{-th FFN expert}}, \tag{4}$$

BTT-MoE learns an MoE in every single linear layer of the model (except the language modeling head) and treats them equally, including the projection matrices $\mathbf{W^Q}, \mathbf{W^K}, \mathbf{W^V}, \mathbf{W^O}$ in the attention blocks [24]. It learns more fine-grained routing decisions among the experts, with $\left(\frac{1}{2}E(E-1)\right)^{6M}$ possible combinations of the experts in a transformer with $M$ blocks, compared to $\left(\frac{1}{2}E(E-1)\right)^{M}$ for the standard MoE architecture.

## 5.2 Compute Efficiency Gains

In Figure 6, we show GPT-2 with BTT-MoE achieves better compute-optimal scaling laws compared to the dense model as well as the standard MoE, with $k = 2$ and $E \in \{8, 16\}$. BTT-MoE consistently outperforms the standard MoE and the dense model. We quantify and compare the compute efficiency gains of BTT-MoE and standard MoE over dense models via the *compute multiplier*. A model with a compute multiplier of $\lambda$ means with $C$ training FLOPs it achieves the same loss as a dense model with $\lambda C$ training FLOPs. In Figure 6, we show BTT-MoE is significantly more compute-efficient than the standard MoE for both $E = 8$ and $E = 16$. In particular, with $E = 16$ experts, BTT-MoE achieves a compute multiplier of $\lambda = 5.3_{\pm 0.3}$, compared to $\lambda = 4.1_{\pm 0.3}$ for a standard MoE.

## 5.3 Effect of Structures

In Figure 7 we show that replacing BTT-MoE with a sparse ($k = 2$) sum of low-rank matrices (low-rank-MoE) or dense matrices (dense-MoE) also yields a nontrivial compute multiplier ($\sim 2\times$) over the dense model, but is significantly less effective than BTT-MoE or even the standard MoE.

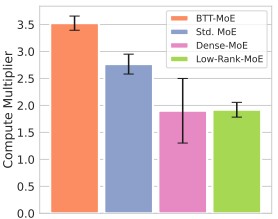

While the poor relative performance of low-rank-MoE is expected, this result shows that in addition to $\omega = 0$ and $\psi = 1$, $\nu < 1$ is a desirable property for the base structure in a structured MoE. Using a dense structure with $\nu = 1$ means the experts are not complementary to each other since each one is able to represent the entire space of dense matrices.

Figure 7: **Mean and std dev of compute multipliers for structured MoE. BTT is better than low rank or dense.**

## 6 Scaling Optimization for Einsums

As prior work [19] has shown, the optimal initialization scales and learning rate depend heavily on the structure of the linear layers and are critical for successfully training models with structured layers. Fortunately, the theory of the Maximal Update Parameterization ($\mu$P) [26–28] and its application

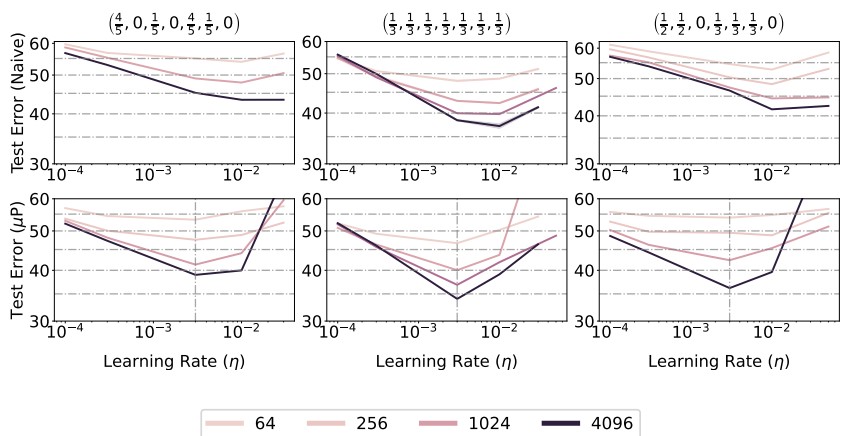

Figure 8: **Einsums trained with $\mu$P achieve lower error and share an optimal base learning rate.** We plot test error of 4 layered MLP models on CIFAR-10, where the hidden layers are Einsums. We vary the model widths in 64, 256, 1024 and 4096. The naive approach uses a global learning rate independent of width or structure and initializes the Einsums parameters with unit variance.

to various structured matrices in Qiu et al. [19] provides a template on how to reason about the appropriate initialization and learning rate scales for all Einsums.

In short, $\mu$P states that for a dense matrix $\mathbf{W} \in \mathbb{R}^{d_{\text{in}} \times d_{\text{out}}}$, the optimal initialization standard deviation scales as $\sigma = \Theta(\sqrt{\min(d_{\text{in}}, d_{\text{out}})/d_{\text{in}}^2})$ and its learning rate as $\eta = \Theta(1/d_{\text{in}})$ if using Adam. Furthermore, Qiu et al. [19] shows we can apply $\mu$P to structured matrices as long as we can cast the MVM as a series of batched matrix multiplications (BMM). As shown in Appendix C, we can indeed cast any Einsum as a series of BMMs and show that $d_{\text{in}}^{\mathbf{A}} = d_{\mathbf{XA}}, d_{\text{out}}^{\mathbf{A}} = d_{\mathbf{YA}} d_{\mathbf{YAB}} d_{\mathbf{AB}}$, $d_{\text{in}}^{\mathbf{B}} = d_{\mathbf{XB}} d_{\mathbf{XAB}} d_{\mathbf{AB}}$ and $d_{\text{out}}^{\mathbf{B}} = d_{\mathbf{YB}}$. As a result, we can compute the optimal scaling of $\sigma_{\mathbf{A}}, \sigma_{\mathbf{B}}, \eta^{\mathbf{A}}$, and $\eta^{\mathbf{B}}$. In particular, for Adam we have $\eta^{\mathbf{A}} = \Theta(\frac{1}{d_{\mathbf{XA}}})$ and $\eta^{\mathbf{B}} = \Theta(\frac{1}{d_{\mathbf{XB}} d_{\mathbf{XAB}} d_{\mathbf{AB}}})$. Figure 8 shows using $\mu$P leads to a stable optimal learning rate and better performance compared to naively using a constant global learning rate and unit initialization variance. This property allows us to transfer the learning rate between structures and model sizes, saving substantial compute for hyperparameter tuning. For $\mu$P, the learning rate refers to that used by a dense model with width 64, which we transfer to the Einsum models of different widths and structures via the scaling rule identified earlier (see details in Appendix C).

Finally, we discuss in Appendix C an alternative way to reason about the optimal learning rates of Einsums via Riemannian SGD (RSGD) [3]. We analyze the effective learning rate prescribed by RSGD at initialization for asymptotically large Einsums and find it often agrees with the $\mu$P prescription derived above.

## 7 Conclusion

Going beyond prior works that study hand-crafted structured matrices on a case-by-case basis, we introduce a continuous parameterization over the space of all structured matrices expressible as Einsums. Using this parameterization, we measure and compare the compute-optimal scaling laws of a wide range of known and novel structures, with the following key takeaways:

- *Compute-optimal scaling laws of Einsums are primarily governed by the parameter-sharing exponent $\omega$ and the rank exponent $\psi$.* Across tasks, we find all full-rank Einsums without parameter sharing (i.e. full-rank BTTs) scale similar to dense, while the remaining vast majority of Einsums consistently underperform dense as $\omega$ increases or $\psi$ decreases.

- *Existing structured matrices do not significantly outperform dense in the compute-optimal setting.* While low rank, Tensor-Train, Monarch, and BTT have shown advantages over dense in other settings, such as controlling for memory or model size, they generally perform worse or similar to dense when controlling for training compute. However, there are also instances in the compute-optimal regime where a full-rank structured representation with no parameter sharing can outperform dense layers. This advantage is most likely due to the ability to make wider structured layers for the same computational budget as narrower dense layers, which can particularly benefit smaller vision models, as we show on CIFAR-5M.

- *$\mu$P prescribes effective initialization and learning rate scaling for Einsums.* Breaking an Einsum down to a sequence of batched matrix multiplications, we extend prior work on structure-aware initialization and learning rate based on $\mu$P to arbitrary Einsums.

- *MoE over structured matrices is more efficient than standard MoE over entire FFNs.* By replacing every single dense linear layer with a sparse sum of structured matrices like BTT, compared to standard MoE which operates over entire FFNs, we create a more efficient MoE architecture, achieving over $5\times$ savings in compute on language modeling relative to dense. Scaling and improving the proposed structured MoE architecture are exciting directions for future work.

**Acknowledgements:** We thank Alan Amin, Nate Gruver, and Hoang Phan for helpful discussions. This work is supported by NSF CAREER IIS-2145492, NSF CDS&E-MSS 2134216, NSF HDR-2118310, BigHat Biosciences, Capital One, and an Amazon Research Award.

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

## Appendix Outline

## A    Examples of Einsums

We explicitly show the expression for some common structures expressed as Einsums. Any index not appearing in the expression has a size of 1 and is thus omitted.

**Dense**    A one-factor Einsum would have been enough to represent dense, though one way to represent it within a two-factor Einsum is as:

$$Y_\phi = \sum_{\gamma\phi} B_{\gamma\phi} A_{\gamma\phi} X_\gamma, \tag{5}$$

which is an over-parameterization due to the Hadamard product $B_{\gamma\phi} A_{\gamma\phi}$.

**Low Rank**

$$Y_\epsilon = \sum_{\alpha\phi} B_{\epsilon\rho} A_{\rho\alpha} X_\alpha, \tag{6}$$

**Kronecker Product**

$$Y_{\delta\epsilon} = \sum_{\alpha\beta} B_{\beta\epsilon} A_{\alpha\delta} X_{\alpha\beta}, \tag{7}$$

**Tensor-Train**

$$Y_{\delta\epsilon} = \sum_{\alpha\beta\rho} B_{\beta\epsilon\rho} A_{\alpha\delta\rho} X_{\alpha\beta}, \tag{8}$$

**Monarch**

$$Y_{\epsilon\phi} = \sum_{\alpha\gamma} B_{\gamma\epsilon\phi} A_{\alpha\gamma\phi} X_{\alpha\gamma} \tag{9}$$

**Block Tensor-Train**

$$Y_{\epsilon\phi} = \sum_{\alpha\gamma\rho} B_{\gamma\epsilon\phi\rho} A_{\alpha\gamma\phi\rho} X_{\alpha\gamma} \tag{10}$$

## B    Taxonomy Derivation

Here we present the derivation for the taxonomy variables $(\omega, \psi, \nu)$. It is helpful to first reason about the amount of compute required for calculating an Einsum, which will allow us to exclude some uninteresting Einsums from consideration and thereby simplify our analysis. We only consider two-factor Einsums representing a square matrix, but generalization to more factors and non-square matrices is straightforward.

For reference, we show the general expression again:

$$Y_{\delta\epsilon\phi} = \sum_{\alpha\beta\gamma\rho} B_{\beta\gamma\epsilon\phi\rho} A_{\alpha\gamma\delta\phi\rho} X_{\alpha\beta\gamma} \tag{11}$$

**Computational Complexity.** There are three possible ways to compute Equation (11), depending on whether the first contraction happens between $\{\mathbf{A}, \mathbf{B}\}$, $\{\mathbf{A}, \mathbf{X}\}$, or $\{\mathbf{B}, \mathbf{X}\}$. The required FLOPs [†] are $\Theta\!\left(d^{2+\theta_\rho}\right)$, $\Theta\!\left(d^{2+\theta_\rho-\min(\theta_\alpha,\theta_\epsilon)}\right)$, and $\Theta\!\left(d^{2+\theta_\rho-\min(\theta_\beta,\theta_\delta)}\right)$ respectively. The optimal computational path depends on the entries of $\boldsymbol{\theta}$, and the optimal FLOPs $F$ required is given by $F = \Theta\!\left(d^{2+\theta_\rho-\max\{\min(\theta_\alpha,\theta_\epsilon),\min(\theta_\beta,\theta_\delta)\}}\right)$.

**Removing the Exchange Redundancy.** The factors $\mathbf{A}$ and $\mathbf{B}$ play an equivalent role in the Einsum expression, so each distinct structure is represented by two vectors that correspond to relabelling $\mathbf{A}$ and $\mathbf{B}$ to each other. To remove this redundancy, we can require that first summing with $\mathbf{A}$ is more computationally efficient. Thus, we require

$$\min(\theta_\alpha, \theta_\epsilon) \geq \min(\theta_\beta, \theta_\delta), \tag{12}$$

which also simplifies the FLOPs to $F = \Theta\!\left(d^{2+\theta_\rho-\min(\theta_\alpha,\theta_\epsilon)}\right)$. To be more exact, we have $F = \rho d^2 \left(\frac{1}{d_\alpha} + \frac{1}{d_\epsilon}\right)$.

**Degenerate Einsums.** Since the overall Einsum is a linear operator on $\mathbb{R}^d$, any Einsum that requires more than $\Theta\!\left(d^2\right)$ FLOPs to compute is degenerate in the sense that unnecessary computations are performed. For example, one such Einsum could correspond to a factorization $\mathbf{U}\mathbf{V}^\intercal$, $\mathbf{U} \in \mathbb{R}^{d\times r}$, $\mathbf{V} \in \mathbb{R}^{r\times d}$ where $r \gg d$. For convenience, we will also define an Einsum whose cost is equal to $\Theta\!\left(d^2\right)$ as degenerate since it is no more efficient than a dense matrix. Given the expression for FLOPs, we conclude that non-degenerate Einsum are those where

$$\theta_\rho < \min(\theta_\alpha, \theta_\epsilon). \tag{13}$$

Intuitively, this requirement means that the rank dimension $d_\rho$, i.e. the range of the index $\rho$ connecting $\mathbf{A}$ and $\mathbf{B}$, cannot be set too high, since after some point it becomes more efficient to simply use a dense matrix in place of the Einsum. In particular, we must have $d_\alpha > 1$ and $d_\epsilon > 1$ (except when $d = 1$).

**Compute Intensity Exponent, $\nu$.** One defining characteristic of structured matrices is that the FLOPs $F$ for performing matrix-vector-multiplication (MVM) is sub-quadratic in $d$. Equivalently, the compute intensity $F/d$, i.e. FLOPs for an MVM normalized by the dimension, is sublinear. In our case, all non-degenerate Einsum have sublinear compute intensity. More precisely, we have $F/d = \Theta(d^\nu)$, where $\nu = 1 + \theta_\rho - \min(\theta_\alpha, \theta_\epsilon)$ takes value in $[0, 1)$ assuming non-degeneracy. The closer $\nu$ is to 1, the more a Einsum resembles a dense matrix and vice versa. $\nu = 0$ corresponds to low rank matrices with $\Theta(1)$ rank.

**Rank Exponent, $\psi$.** As we show in Appendix C.1, the Einsum can be computed via two batched matrix multiplications where $\mathbf{A}$ acts as a matrix consisting of $d_\beta d_\gamma$ blocks of $d_\alpha \times d_\delta d_\phi d_\rho$ matrices. Thus, $\mathrm{rank}(\mathbf{A}) = \min(d, d_\beta d_\gamma d_\delta d_\phi d_\rho) = d^{\min(1, \theta_\beta + \theta_\gamma + \theta_\delta + \theta_\phi + \theta_\rho)}$ as $d_\alpha d_\beta d_\gamma = d$. Similarly, we have that $\mathbf{B}$ acting as a matrix consisting of $d_\delta d_\phi$ blocks of $d_\beta d_\gamma d_\rho \times d_\epsilon$ matrices. Therefore we have that $\mathrm{rank}(\mathbf{B}) = d^{\min(1, \theta_\beta + \theta_\gamma + \theta_\delta + \theta_\phi + \theta_\rho)}$. Since $\mathbf{W}$ is the product of $\mathbf{A}, \mathbf{B}$, and some reshape operations (which are full-rank), $\mathrm{rank}(\mathbf{W}) = d^\psi$ is the minimum of $\mathrm{rank}(\mathbf{A})$ and $\mathrm{rank}(\mathbf{B})$ :

$$\begin{aligned}
\psi &= \min(1, \theta_\beta + \theta_\gamma + \theta_\delta + \theta_\phi + \theta_\rho) \\
&= \min(1, 2 + \theta_\rho - \theta_\alpha - \theta_\epsilon,).
\end{aligned}$$

Note that, technically, when we write $\mathrm{rank}(\mathbf{M}), \mathbf{M} \in \{\mathbf{A}, \mathbf{B}, \mathbf{W}\}$, we mean the maximum possible rank of $\mathbf{M}$ when its parameters are learned. Otherwise the equalities will become upper-bounds. For low-rank we have $\boldsymbol{\theta} = (1, 0, 0, 0, 1, 0, \theta_\rho)$ and hence $\psi = \theta_\rho$. For dense we have $\boldsymbol{\theta} = (0, 0, 1, 0, 0, 1, 0)$ and hence $\psi = 1$. For Kronecker we have $\boldsymbol{\theta} = (1/2, 1/2, 0, 1/2, 1/2, 0, 0)$ and hence $\psi = 1$. For BTT we have $\psi = \min(1, \theta_\gamma + \theta_\phi + \theta_\rho)$.

---

[†]We use the more familiar term FLOPs as a stand-in for MACs (Multiply-Accumulate), even though 1 MAC is technically 2 FLOPs.

**Parameter Sharing Exponent** $\omega$. The number of learnable parameters $N$ in a Einsum is simply the sum of the elements in $\mathbf{A}$ and $\mathbf{B}$, which works out to be $N = d_\rho d^2 \left( \frac{1}{d_\alpha d_\delta} + \frac{1}{d_\beta d_\epsilon} \right)$. Since the FLOPs is $F = d_\rho d^2 \left( \frac{1}{d_\alpha} + \frac{1}{d_\epsilon} \right)$, only Einsums with $d_\delta = d_\beta = 1$, or equivalently $\theta_\delta = \theta_\beta = 0$, have parameters matching FLOPs. In general, the number of parameters per FLOP $N/F = \Theta(d^{-\omega})$ with $\omega = \min(\theta_\alpha + \theta_\delta, \theta_\beta + \theta_\epsilon) - \min(\theta_\alpha, \theta_\epsilon) \geq 0$.

The $\omega = 0$ subspace is of particular interest because any Einsum outside of this subspace has an artificially limited expressivity per FLOP because it reuses each parameter multiple times. In this sense, the $\omega = 0$ subspace is the space of maximally expressive Einsums that maximizes expressivity per FLOP. We note that this subspace precisely corresponds to the Block Tensor-Train (BTT) structure proposed in Qiu et al. [19], provided we allow a minor generalization of the original BTT structure so that an axis connects the first factor back to the last factor when there are more than 2 factors, exactly analogous to the generalization of the Tensor-Train structure to the Tensor Ring [30] structure.

## C    Scaling Optimization for Einsums

### C.1    Learning Rate Scaling

In Section 6, we claimed that the Adam learning rates should be scaled as $\eta^{\mathbf{A}} = \Theta(\frac{1}{d_{\mathbf{XA}}})$ and $\eta^{\mathbf{B}} = \Theta(\frac{1}{d_{\mathbf{XB}} d_{\mathbf{XAB}} d_{\mathbf{AB}}})$. Arriving at these scaling rules requires (1) expressing an Einsum as a sequence of batched matrix multiplies (BMMs) and showing $d_{\text{in}}^{\mathbf{A}} = d_{\mathbf{XA}}, d_{\text{out}}^{\mathbf{A}} = d_{\mathbf{YA}} d_{\mathbf{YAB}} d_{\mathbf{AB}}$, $d_{\text{in}}^{\mathbf{B}} = d_{\mathbf{XB}} d_{\mathbf{XAB}} d_{\mathbf{AB}}$ and $d_{\text{out}}^{\mathbf{B}} = d_{\mathbf{YB}}$, and (2) applying results from Qiu et al. [19] on learning rate scaling for structured matrices expressible in terms of BMMs. We now justify (1).

The general 2-factor Einsum

$$Y_{\delta\epsilon\phi} = \sum_{\alpha\beta\gamma\rho} B_{\beta\gamma\epsilon\phi\rho} A_{\alpha\gamma\delta\phi\rho} X_{\alpha\beta\gamma} \tag{14}$$

can be computed in two steps. In step 1,

$$Z_{\beta\gamma\delta\phi\rho} = \sum_\alpha A_{\alpha\gamma\delta\phi\rho} X_{\alpha\beta\gamma}, \tag{15}$$

which is a BMM with $\beta\gamma$ acting as batch dimensions, $\alpha$ the input dimension, and $\delta\phi\rho$ the output dimensions. So $d_{\text{in}}^{\mathbf{A}} = d_{\mathbf{XA}}, d_{\text{out}}^{\mathbf{A}} = d_{\mathbf{YA}} d_{\mathbf{YAB}} d_{\mathbf{AB}}$. In step 2,

$$Y_{\delta\epsilon\phi} = \sum_{\beta\gamma\rho} B_{\beta\gamma\epsilon\phi\rho} Z_{\beta\gamma\delta\phi\rho}, \tag{16}$$

which is a BMM with $\delta\phi$ acting batch dimensions, $\beta\gamma\rho$ input dimensions, and $\epsilon$ the output dimension. So $d_{\text{in}}^{\mathbf{B}} = d_{\mathbf{XB}} d_{\mathbf{XAB}} d_{\mathbf{AB}}$ and $d_{\text{out}}^{\mathbf{B}} = d_{\mathbf{YB}}$, as wanted.

In our experiments, we apply the above scaling rule to transfer from a learning rate $\eta$ used by a dense matrix with width $d_0$ to learning rates for $\mathbf{A}, \mathbf{B}$ as

$$\eta^{\mathbf{A}} = \frac{d_0}{2d_{\mathbf{XA}}}\eta, \quad \eta^{\mathbf{B}} = \frac{d_0}{2d_{\mathbf{XB}} d_{\mathbf{XAB}} d_{\mathbf{AB}}}\eta, \tag{17}$$

where the additional factor of two is to account for both $\mathbf{A}$ and $\mathbf{B}$ contributing updates to the output of each layer, following Qiu et al. [19].

### C.2    Connections between $\mu$P and Riemannian SGD

Riemannian SGD (RSGD) is an optimization technique that allows us to perform the equivalent of SGD on a Riemannian manifold consisting of points $\{\mathbf{q}\}$ with a metric $g_{\mathbf{q}}$ [3]. The updates of RSGD is almost identical to SGD: $\mathbf{q}^{(t+1)} = \mathbf{q}^{(t)} - \eta_{\mathbf{q}} \, g_{\mathbf{q}^{(t)}}^{-1} \nabla_{\mathbf{q}} \mathcal{L}(\mathbf{q}^{(t)})$, except that the gradient is multiplied by the inverse of the metric $g_{\mathbf{q}}$. In our case, we want to mimic training the Einsum parameters $\mathbf{A}$ and $\mathbf{B}$ as if we were training $\mathbf{W}$ with SGD directly, even though we will never represent $\mathbf{W}$ explicitly. Thus, we identify (flattened) $(\mathbf{A}, \mathbf{B})$ with $\mathbf{q}$ and specify its metric as the pull-back metric of the

Euclidean metric $g_{\mathbf{W}} = \mathbf{I}$ on $\mathbf{W}$, which is given by $g_{\mathbf{q}} = \mathbf{J}(\mathbf{q})^{\mathsf{T}}\mathbf{J}(\mathbf{q})$ where $\mathbf{J}(\mathbf{q}) = \frac{\partial \mathbf{W}}{\partial \mathbf{q}}$. The RSGD updates to $\mathbf{q}$ is therefore

$$\mathbf{q}^{(t+1)} = \mathbf{q}^{(t)} - \eta_{\mathbf{q}} \left( \mathbf{J}(\mathbf{q}^{(t)})^{\mathsf{T}} \mathbf{J}(\mathbf{q}^{(t)}) \right)^{-1} \nabla_{\mathbf{q}} \mathcal{L}(\mathbf{q}^{(t)}).$$

Exactly computing the inverse metric would require $O(P^3)$ time, where $P$ is the number of parameters in $\mathbf{A}, \mathbf{B}$. It is, therefore, too expensive to run RSGD during training. However, we can efficiently approximate the inverse metric at initialization in a way that is exact for asymptotically large matrices. We compute $\mathbf{J}^{\mathsf{T}}\mathbf{J}$ through three blocks $\mathbf{J}_{\mathbf{A}}^{\mathsf{T}}\mathbf{J}_{\mathbf{A}}$, $\mathbf{J}_{\mathbf{B}}^{\mathsf{T}}\mathbf{J}_{\mathbf{B}}$, and $\mathbf{J}_{\mathbf{A}}^{\mathsf{T}}\mathbf{J}_{\mathbf{B}}$ and note that for a fixed $\theta$ as we increase both $d_{\mathrm{in}}$ and $d_{\mathrm{out}}$, we have that $\mathbf{J}_{\mathbf{A}}^{\mathsf{T}}\mathbf{J}_{\mathbf{A}} \approx d_{\beta}d_{\epsilon}\sigma_{\mathbf{B}}^2\mathbf{I}$, $\mathbf{J}_{\mathbf{B}}^{\mathsf{T}}\mathbf{J}_{\mathbf{B}} \approx d_{\alpha}d_{\delta}\sigma_{\mathbf{A}}^2\mathbf{I}$, and $\mathbf{J}_{\mathbf{A}}^{\mathsf{T}}\mathbf{J}_{\mathbf{B}} \approx \mathbf{0}$ where $\sigma_{\mathbf{A}}$ and $\sigma_{\mathbf{B}}$ denote the initialization scales of $\mathbf{A}$ and $\mathbf{B}$. Therefore we have

$$(\mathbf{J}^{\mathsf{T}}\mathbf{J})^{-1} \approx \begin{pmatrix} \mathbf{I}/(d_{\beta}d_{\epsilon}\sigma_{\mathbf{B}}^2) & \mathbf{0} \\ \mathbf{0} & \mathbf{I}/(d_{\alpha}d_{\delta}\sigma_{\mathbf{A}}^2) \end{pmatrix}.$$

We can now identify $\Theta\big(1/(d_{\beta}d_{\epsilon}\sigma_{\mathbf{B}}^2)\big)$ as $\eta_{\mathbf{A}}$ and $\Theta\big(1/(d_{\alpha}d_{\delta}\sigma_{\mathbf{A}}^2)\big)$ as $\eta_{\mathbf{B}}$. If we further use the initialization scales prescribed by $\mu$P as given in Section 6, we have

$$\eta_{\mathrm{RSGD}}^{\mathbf{A}} = \Theta\left( \frac{1}{d_{\beta}d_{\epsilon}} \frac{d_{\beta}^2 d_{\gamma}^2 d_{\rho}^2}{\min(d_{\epsilon}, d_{\beta}d_{\gamma}d_{\rho})} \right) \quad \text{and} \quad \eta_{\mathrm{RSGD}}^{\mathbf{B}} = \Theta\left( \frac{1}{d_{\alpha}d_{\delta}} \frac{d_{\alpha}^2}{\min(d_{\alpha}, d_{\delta}d_{\phi}d_{\rho})} \right).$$

It is now interesting to compare these scalings to the prescriptions of $\mu$P SGD. For SGD, $\mu$P proposes to scale the learning rate as $\Theta(d_{\mathrm{out}}/d_{\mathrm{in}})$ [29] for dense matrices. Unlike in Adam, to extend $\mu$P to Einsums, we need not only to replace $d_{\mathrm{in}}, d_{\mathrm{out}}$ with $d_{\mathrm{in}}^{\mathbf{A}}, d_{\mathrm{out}}^{\mathbf{A}}$, or $d_{\mathrm{in}}^{\mathbf{B}}$ and $d_{\mathrm{out}}^{\mathbf{B}}$, but also scale down the learning rate as $\Theta(1/d_{\beta})$ for $\mathbf{A}$ and $\Theta(1/d_{\delta})$ for $\mathbf{B}$. The final $\mu$P learning rates have two possibilities depending on whether we consider $\mathbf{A}$ or $\mathbf{B}$ as the first layer. If we consider $\mathbf{A}$ as the first layer, then

$$\eta_{\mu\mathrm{P}}^{\mathbf{A}} = \Theta\left( \frac{1}{d_{\beta}} \frac{d_{\beta}d_{\gamma}d_{\rho}}{d_{\alpha}} \right) \quad \text{and} \quad \eta_{\mu\mathrm{P}}^{\mathbf{B}} = \Theta\left( \frac{1}{d_{\delta}} \frac{d_{\epsilon}}{d_{\delta}d_{\phi}d_{\rho}} \right).$$

Evidently, the two approaches don't always agree. However, they are identical when $d_{\alpha} = d_{\epsilon}, d_{\beta}d_{\gamma}d_{\rho} \leq d_{\epsilon}$, and $d_{\delta}d_{\phi}d_{\rho} \leq d_{\alpha}$. In fact, many structures satisfy this condition. If we assume the structure is symmetric, meaning its transpose can be represented with the same $\theta$, i.e. $d_{\alpha} = d_{\epsilon}, d_{\beta} = d_{\delta}$, and $d_{\gamma} = d_{\phi}$, then the remaining conditions simplify to only $d_{\rho} \leq \frac{d_{\alpha}^2}{d}$. Therefore, any symmetric Einsum with $\theta_{\alpha} \geq 1/2$ and $\theta_{\rho} \leq 2\theta_{\alpha} - 1$ satisfy these conditions. Thus, for these structures, $\mu$P SGD can be viewed as an approximation to RSGD that is valid at initialization in the infinite-width limit. While we cannot establish a direct connection between RSGD and $\mu$P Adam, which is what we use in our experiments and broadly in large-scale training, $\mu$P Adam is similar to $\mu$P SGD in that it maximizes feature learning per layer [27]. The connection between $\mu$P SGD and RSGD therefore indirectly provides an alternative justification for $\mu$P Adam.

## D    Experiments

### D.1    GPT-2 with Original Vocabulary and Longer Context

In Figure 9, we show our findings in Section 4 translate to the more standard GPT-2 evaluation with a longer sequence length of 512 and its original vocabulary of 50,257 tokens. We train models with $L = 12$ layers up to the GPT-2 Small [20] size by increasing width $d$. We use Adam with a base learning rate of 0.002 for a $L = 3, d = 256$ dense model, which is scaled to other models via $\mu$P. Since the language modeling head contains a significant fraction of the parameters for models of this scale, we replace all layers, including the head, with Einsums.

Qualitatively, Figure 9 differs from Figure 4 in two ways: 1) the scaling laws are less power law like and show some curvature on a log-log scale, and 2) BTT with $\nu > 0$ seems to perform better than $\nu = 0$. We believe 1) is due to the increased context length and vocabulary size, making the loss no longer follow a clean power law at the small scales we tested [13, 10]. This was an important motivation for performing experiments with a smaller vocabulary size and context length in Section 4. Similarly, we believe the increased vocabulary size and context length contributed to 2), as a larger $\nu$ implies at small scales a higher fraction of compute are in the transformer blocks rather than the language modeling head, which likely improves performance. By contrast, in our setup in Section 4, the model dimension $d$ dominates the vocabulary size and context length, leading to less significant finite-size effects.

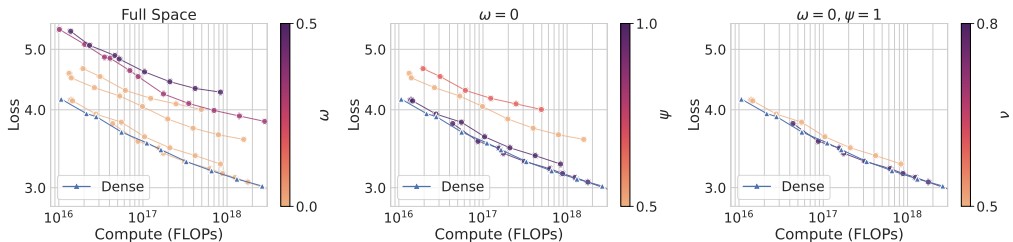

Figure 9: **The taxonomy parameters** $(\omega, \psi)$ **explain differences in the Einsum scaling laws for 12-layer GPT-2 models with standard vocabulary (50,257 tokens) and a context length of 512.** Small $\omega$ (no parameter sharing) and large $\psi$ (full-rank) are necessary for a structure to perform well, while variation in $\nu$ has a much smaller impact on performance.

### D.2 Autoregressive Pixel Modeling on CIFAR-5M

We train 2 and 3 layer transformers with Adam using a base learning rate of 3e-3 for a width 64 dense model. The width ranges from 32 to 512 for dense and 32 to 1024 for Einsums. All models are trained for 2 epochs with a batch size of 64. We subtract an estimated irreducible loss of 3.25 before reporting the loss in Figure 5 (top row).

### D.3 MLP Regression on Synthetic Data.

We train 3-layer MLP models with width $d \in [64, 4096]$ for a maximum of $10^6$ steps and a batch size of 4096 on an effectively infinite synthetic dataset. The synthetic dataset is obtained by querying a scalar-valued target function on $\mathbb{R}^8$ with inputs drawn from a Gaussian distribution. The target function is a randomly initialized target MLP with 6 layers and a hidden dimension of 1024. We minimize the Mean-Squared-Error (MSE) loss. We train with Adam using a base learning rate of 1e-3 for a width 64 dense model. We report the raw MSE loss in Figure 5 (bottom row).

## E   Generalization to More than Two Factors

The generalization to more factors is easy to understand if we consider the set of edges that define the Einsum $\mathcal{E} = \{S \subseteq \{\mathbf{X}, \mathbf{A}, \mathbf{B}, \mathbf{Y}\} : |S| \geq 2 \text{ and } \{\mathbf{X}, \mathbf{Y}\} \not\subseteq S\}$. For example, for three factors, the edges that connect to $\mathbf{X}$ are $\{\mathbf{X}, \mathbf{A}\} \leftrightarrow i_1, \{\mathbf{X}, \mathbf{B}\} \leftrightarrow i_2, \{\mathbf{X}, \mathbf{C}\} \leftrightarrow i_3, \{\mathbf{X}, \mathbf{A}, \mathbf{B}\} \leftrightarrow i_4, \{\mathbf{X}, \mathbf{B}, \mathbf{C}\} \leftrightarrow i_5 \{\mathbf{X}, \mathbf{A}, \mathbf{C}\} \leftrightarrow i_6$ and $\{\mathbf{X}, \mathbf{A}, \mathbf{B}, \mathbf{C}\} \leftrightarrow i_7$. The edges that connect to $\mathbf{Y}$ are $\{\mathbf{Y}, \mathbf{A}\} \leftrightarrow j_1, \{\mathbf{Y}, \mathbf{B}\} \leftrightarrow j_2, \{\mathbf{Y}, \mathbf{C}\} \leftrightarrow j_3, \{\mathbf{Y}, \mathbf{A}, \mathbf{B}\} \leftrightarrow j_4, \{\mathbf{Y}, \mathbf{B}, \mathbf{C}\} \leftrightarrow j_5 \{\mathbf{Y}, \mathbf{A}, \mathbf{C}\} \leftrightarrow j_6$ and $\{\mathbf{Y}, \mathbf{A}, \mathbf{B}, \mathbf{C}\} \leftrightarrow j_7$. The edges between the factors are $\{\mathbf{A}, \mathbf{B}\} \leftrightarrow r_1, \{\mathbf{A}, \mathbf{C}\} \leftrightarrow r_2$ and $\{\mathbf{A}, \mathbf{B}, \mathbf{C}\} \leftrightarrow r_3$. The expression for a three factor would be

$$Y_{\mathbf{j}} = \sum_{\mathbf{i}, \mathbf{r}} A_{i_1 i_4 i_6 i_7 j_1 j_4 j_6 j_7 r_1 r_3} B_{i_2 i_4 i_5 i_7 j_2 j_4 j_5 j_7 r_1 r_3} C_{i_3 i_5 i_6 i_7 j_3 j_5 j_6 j_7 r_2 r_3} X_{\mathbf{i}}.$$

For more factors we follow the combinatorial procedure of listing the sets and adding an index based on each edge in $\mathcal{E}$.

## F   Exploiting the Attention Structure

In self-attention, given an embedding vector $\mathbf{x} \in \mathbb{R}^d$, we compute $\mathbf{q} = \mathbf{W^Q x}$, $\mathbf{v} = \mathbf{W^V x}$ and $\mathbf{k} = \mathbf{W^K x}$. After computing each of the $\mathbf{q}, \mathbf{k}, \mathbf{v} \in \mathbb{R}^d$ vectors, they are reshaped to produce one smaller vector per attention head: $q_i \rightarrow q_{hj}, k_i \rightarrow k_{hj}, v_i \rightarrow v_{hj}$ where $h$ is an axis of size $H$, the number of attention heads, and $j$ is an axis of size $d/H$. When replacing $\mathbf{W^Q}, \mathbf{W^K}, \mathbf{W^V}$ with BTTs, it is therefore more natural to have the BTT output axes $\epsilon\phi$ coincide with $hj$ so the MVM is aware of the attention head structure rather than potentially mixing different heads. Similarly, when replacing $\mathbf{W^O}$ with a BTT, it is most natural to have the BTT input axes

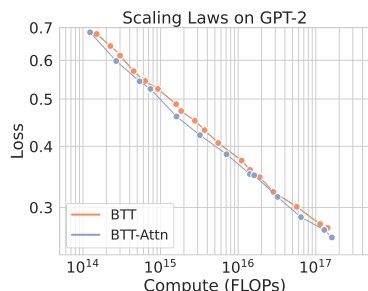

Figure 10: **Exploiting attention head structure improves compute-efficiency by an average of 17%.**

$\alpha\gamma$ coincide with $hj$. In Figure 10, we show doing so slightly improves compute efficiency by an average of 17% over naively replacing all attention and FFN matrices with BTT, which corresponds to $\boldsymbol{\theta} = (1/2, 0, 1/2, 0, 1/2, 1/2, 0)$ in Section 4.

## G   Hardware specifications

Our experiments in Section 4 are run on on A100 and H100 GPUs. The CIFAR-10 experiments in Section 6 were run on RTX2080 Ti and RTX Titan GPUs.

