# OpenReview forum: "Searching for Efficient Linear Layers over a Continuous Space of Structured Matrices"
_NeurIPS.cc/2024/Conference — NeurIPS 2024 poster_

### Official Review · Reviewer_XAQT · 2024-07-09

**Soundness:** 4
**Presentation:** 2
**Contribution:** 4
**Rating:** 7
**Confidence:** 5

**Summary:**

The paper generalizes multiple existing structured matrices by means of Einsum. The scaling law of the structured matrices with different rank, compute intensity, and parameters-per-FLOPs is analyzed on GPT-2. Since the high-rank, non-parameter-sharing einsum operations obtain the best results, the paper proposes a sparse mixture of structured linear experts which also has high-rank, non-parameter-sharing taxonomy parameters with further generalization. Additionally, the learning rate scaling rule for the Einsum-induced structured matrices via maximal update parameterization is introduced.

**Strengths:**

- The paper provides a novel point of view for understanding the structured matrices via Einsum operations.
- The paper is clearly written with sufficient details.
- The comprehensive analysis of the interesting aspects of the structured matrices leaves valuable insights for future research.

**Weaknesses:**

- Although the text is well written, it was hard to keep track of a dozen of alphabets indicating dimensions, taxonomy variables, etc. Reminding the role of each alphabet occasionally would improve the readability.
- The continuity of the taxonomy space is questionable. It seems like the taxonomy space is discrete because rank, FLOPs, and number of parameters are all integers. Even if they are normalized, the parameters would still reside in a finite sized space of rational numbers from 0 to 1.

**Questions:**

- Is the scaling law of Einsum generalizable to the larger models (e.g., $\ge$7B parameters)?
- Are there any insights or possible reason that the authors think why the high-rank, no parameter sharing leads to the best results whereas the compute intensity does not affect the accuracy?
- Could the taxonomy parameters have any real numbers other than rational numbers?

**Limitations:**

- The framework and the analysis is confined to a macroscopic point of view--the test was conducted upon fixing the einsum configuration and the FLOPs across the layers, whereas the optimal configuration and FLOPs might vary from layer to layer.

---

> ### Author Rebuttal · Authors · 2024-08-07
>
> We sincerely appreciate your thoughtful and supportive review. We agree that the large number of variables presents readability challenges and will update the paper to provide reminders about the meanings of the indices. We now provide several clarifications and new results inspired by your comments.
>
> **On continuity of the parameterization**.
> While quantities including FLOPs, parameter count, and rank are discrete, the continuous parameterization of the space of Einsums with non-negative real-valued coordinates $\theta$ is valid. First, any non-negative $\theta$ satisfying the condition $\theta_{{X}{A}} + \theta_{{X}{B}} + \theta_{{X}{A} {B}} = \theta_{{Y}{A}} + \theta_{{Y}{B}} + \theta_{{Y}{A} {B}} = 1$ produces a valid structure once the resulting weight factors are rounded to have integer sizes. Including irrational entries in $\theta$ is completely allowed. Now it suffices to show that any two such coordinates, say $\theta$ and $\theta + \epsilon$ indeed represent a distinct set of structures as the models scale, for any $\epsilon \neq 0$. This is because for large enough dimensions $d$, any small difference of $\epsilon$ in the coordinates will lead to a larger than 1 difference in the size of the weight factors along some axis, which will persist even after rounding to nearest integers. In other words, we have shown that the space of allowed $\theta$ is real-valued and is a bijection with the space of unique Einsum structures.
>
> **On experiments with larger models**.
> In Figure 1 in the [rebuttal's pdf](https://openreview.net/attachment?id=cH4w74hFGe&name=pdf), inspired by your comments, we show results from new experiments on larger GPT-2 models. We train 12 layered models up to the size of original GPT-2 Small [5], adopting the original vocabulary with 50,257 tokens and using a context length of 512. The results agree with our findings in Section 4 which used a reduced vocabulary of 96 tokens and a shorter context length of 128 to save cost. Due to computational constraints, we cannot experiment with 7B parameter models.
>
> In addition, we perform experiments with other architectures and datasets in Figure 2 and 3 in the [rebuttal's pdf](https://openreview.net/attachment?id=cH4w74hFGe&name=pdf), including Vision Transformers on image generation and MLP on synthetic regression. The results confirm that our findings generalize to much broader settings.
>
> **On a studying per layer configurations.**
> You raise a really interesting and exciting direction for future work. Indeed, a proper exploration of different structures per layer would require a significant amount of compute and, most importantly, to construct a method to explore the combinatorial search space. We believe this is an important next step and these are the type of questions that we are excited to see that our work is raising.
>
> We value your support and thoughtful questions. We put a significant effort into our response and would appreciate it if you could consider increasing your score in light of our replies.

---

> > ### Comment · Reviewer_XAQT · 2024-08-10
> >
> > Thank you for your response. I suggest including the definition of continuity of the taxonomy parameters discussed above in the paper. I will keep my score.

---

### Official Review · Reviewer_bZBq · 2024-07-09

**Soundness:** 2
**Presentation:** 2
**Contribution:** 2
**Rating:** 5
**Confidence:** 3

**Summary:**

This paper proposes a general framework to cover different linear layers by continuous parametrization. The authors conduct extensive experiments to demonstrate several optimal design principles for linear layers, and further propose a novel sparse MoE architecture that improves upon existing works.

**Strengths:**

This paper is clearly written and easy to understand

**Weaknesses:**

- The novelty of this work is not clear enough from my perspective
- Empirical results may need further improvements to better support the proposed method

**Questions:**

- I am first puzzled by the novelty of this work. While the authors have conducted extensive experiments on different architectures, what are the key insights and contributions of this submission? It seems that the idea of combining BTT and (sparse) MoE is novel, but it seems straightforward and the authors may provide some more insights for it.
- While the authors mentioned in section 2 that generalization to more than 2 factors is straightforward, it seems not really the case as the selection of different indexes ($\alpha$, $\beta$, … in (1)) requires manual design to ensure expressive power. As such, the authors may need to provide some examples on using more factors and see how existing methods may be covered by such more expressive frameworks.
- Throughout this paper, I only see experiments on GPT-2, which may not be sufficient to derive general conclusions regarding the scaling laws of different Einsums. The authors may need to consider different model architectures (e.g., BERT, ViT) to better support any conclusions here.
- Moreover, current experiments are only conducted on one data set, which is not sufficient to support such general conclusions in this submission either. The authors should also consider experiments on different data sets to derive general conclusions.
- Also, the comparison of different MoE architectures in Figure 6 may not be enough to support the superiority of BTT-MoE. While Figure 3 and 4 indicate that BTT may be optimal for dense architectures, it may not be directly generalized to MoE architectures. The authors may need to include some other MoE architectures, possibly low-rank-MoE as low-rank performs quite close to BTT from Figure 3.

## Post rebuttal

After checking the rebuttal, I still doubt the claim that solely two parameters $\omega$ (parameter sharing) and $\Psi$ (closeness to full-rank) reliably led to better scaling laws, which motivates the authors to design BTT and combine it with MoE. Despite such weakness, I think the authors have made their contribution clear with sufficient support (additional architectures, datasets and other MoE structures). I have increased my score towards acceptance.

**Limitations:**

This paper does not have direct negativie societal impact from my perspective.

---

> ### Author Rebuttal · Authors · 2024-08-07
>
> We appreciate your feedback. Inspired by your comments, we have run additional experiments and provide clarifications below. We hope that you will consider these new results and clarifications in your final assessment.
>
> **On the key contributions of our work:**
> We highlight that this work provides at least 4 novel and impactful contributions.
>
> 1. We provide a unifying framework to conveniently parameterize a large and continuous space of hardware-efficient structured matrices via Einsums. We show this space contains many popular structured matrices explored in prior works, such as low-rank, Kronecker, Tensor-Train, Monarch, and Block Tensor-Train, while most structures within this space are novel. We further develop an informative taxonomy of the space of Einsums based on key properties relevant to machine learning, including the extent of parameter-sharing, matrix rank, and computational complexity.
>
> 2. We perform the first extensive comparison of compute-optimal scaling laws for structured matrices in language modeling. State-of-the-art large language models (LLMs), including the recent Llama 3.1 405B, are purposely trained to be compute-optimal [1, 2], whereas prior works on training with structured matrices do not compare performance under compute optimality (e.g. by training for too many epochs) [3,4]. Our results therefore provide the more appropriate comparison for evaluating structured matrices in realistic LLM training. Indeed, our results reveal that structures such as Monarch that significantly outperform dense in other contexts at best match dense performance in compute-optimal scaling laws.
>
> 3. We show that differences in the compute-optimal scaling laws across a wide range of structures are mostly governed by a small number of variables defined in our taxonomy. Small $\omega$ (less parameter sharing) and large $\psi$ (closer to full-rank) reliably led to better scaling laws, while $\nu$ (how dense-like a structure is) can be varied while leaving the scaling laws almost unchanged. These insights will make future search for performant structured matrices significantly more efficient than random.
>
> 4. Guided by the insight that full-rank ($\psi=1$) structures that maximize parameters ($\omega=0$) per unit of compute performs the best (as argued in our taxonomy and experiments), we propose BTT-MoE, a novel Mixture-of-Experts (MoE) architecture obtained by sparsifying computation in the BTT structure, proving to be 440% and 28% more compute-efficient than dense and standard MoE for training GPT-2 respectively.
>
> **On experiments with additional architectures and datasets.**
> Focusing on compute-optimal scaling laws limits the range of datasets that we can study and therefore we focused on language, as it is standard in scaling laws literature. However, to address your valid concern we have now incorporated Figure 2 and Figure 3 in the [rebuttal's pdf](https://openreview.net/attachment?id=cH4w74hFGe&name=pdf) where we run new experiments to verify that our findings regarding the relative performance of different Einsums continue to hold in other settings:  1) Vision transformers trained with cross-entropy loss for autoregressive image generation on CIFAR-5M; and 2) MLP trained with Mean-Squared-Error loss on synthetic data generated by a large and randomly initialized MLP. The results demonstrate that our findings in the GPT-2 experiments indeed generalize to other architectures and datasets. We will include these results in the updated paper.
>
> **On comparing against MoE with other structures.**
> Following your suggestion, we compare against two additional MoE architectures in Figure 4 in the [rebuttal's pdf](https://openreview.net/attachment?id=cH4w74hFGe&name=pdf): Dense-MoE and Low-Rank-MoE, which similarly replace all linear layers including those in the attention blocks with an MoE over dense or low-rank matrices. The results show that indeed BTT has a unique advantage over other structures for constructing structured MoEs. We will include these results in the updated paper.
>
> **On generalization to more than 2 factors.**
> The set of indices in Equation 1 does not require manual design to ensure expressive power. Instead, we obtain Equation 1 by simply allowing all possible indices to exist. Following the discussion on Line 92, this set of indices can be directly read off from a graphical representation of the Einsum with each index corresponding to a hyperedge among the input, output, and the weight factors. As a result, generalization to more than $N > 2$ factors follows by constructing a graphical representation of the $N$ factor Einsum, assigning an index to each hyperedge, and writing down the resulting Einsum expression. It is easier to visualize this generalization and therefore we now demonstrate this process for $N=3$ in Figure 4 (c) in the [rebuttal's pdf](https://openreview.net/attachment?id=cH4w74hFGe&name=pdf) and show how the resulting expression covers the general 3-factor case as well as BTT. We will include this new figure for $N=3$ as it visually makes the generalization to more factors apparent
>
> Thank you again for your review. We made a significant effort to address your questions, which has substantially improved our work. We would appreciate it if you would consider raising your score in light of our response.
>
> [1] Dubey et al. 2024. The Llama 3 Herd of Models.
>
> [2] Hoffmann et al. 2022. Training compute-optimal large language models.
>
> [3] Dao et al. 2022. Monarch: Expressive structured matrices for efficient and accurate training
>
> [4] Qiu et al. 2024. Compute Better Spent: Replacing Dense Layers with Structured Matrices

---

### Official Review · Reviewer_Mhte · 2024-07-13

**Soundness:** 2
**Presentation:** 3
**Contribution:** 3
**Rating:** 7
**Confidence:** 3

**Summary:**

In this paper, the authors explore the computational efficiency of various structured layers in language modeling tasks. Specifically, they propose a general parametrization of linear operators and conduct an empirical study on the conditions for scalable decomposition based on three key characteristics: rank, compute intensity, and parameters per flops. Moreover, the authors integrate Mixture-of-Experts (MoE) with structured layers, conducting experiments and comparing the results to those obtained using the standard MoE.

**Strengths:**

I believe that the topic of this paper is quite important and timely. With the increasing size of models, it is crucial to find effective ways to efficiently compress them. Among the various matrix and tensor factorization approaches available for compression, it becomes essential to unify them and determine which aspects make them most effective.

**Weaknesses:**

1) I understand that to check multiple different configurations of layers while having limited computational resources you need to somehow restrict your experiments. But what is lacking in my opinion, is verifying your findings with at least several runs in the proper setting.
2) As far as I understand, classic MoE is applied only to FFN, while you also apply it to all the layers, including Q, K, V. This may affect comparison with MoE. See, for example, in Figure 6.

**Questions:**

1) Did you try other than BTT structured layers in the MoE experiments? Why do you only use BTT?
2) It seems to me that rounding, e.g., $d_{in}^\theta$ to the nearest integer interferes with maintaining the shape $d_{in}$. Is it actually the case and if so, how do you deal with it?
3) Do you expect that all the observations about $\omega, \psi \mu$ will remain the same with bigger dictionary?

---

> ### Author Rebuttal · Authors · 2024-08-07
>
> We appreciate your thoughtful feedback. Indeed, unifying existing structured approaches and performing extensive and well-controlled comparisons between them is an important contribution of this work. We now provide additional results and clarifications to your questions.
>
> **On experiments with larger vocabulary and longer context**.
> Thank you for raising this point as these new results strengthen the paper significantly.
> In Figure 1 in the [rebuttal's pdf](https://openreview.net/attachment?id=cH4w74hFGe&name=pdf), we now train models up to the size of original GPT-2 Small [1], adopting the vocabulary of 50,257 tokens and using a context length of 512. The results agree with our previous findings in Section 4 with a reduced vocabulary and a shorter context, showing that our results indeed hold in more realistic settings. Moreover, in Figure 2 and Figure 3, we evaluate on two additional datasets including image generation with Vision Transformers and synthetic regression with MLPs, further demonstrating the generality of our findings. We will include these results in the updated paper.
>
> **On additional MoE comparisons**.
> Initially we followed the standard approach of only comparing against the MoE that is only applied to the FFN layers. However, to address your concern on the lack of MoE modules in attention layers of the baseline, we now compare with two additional alternatives in Figure 4 in the [rebuttal's pdf](https://openreview.net/attachment?id=cH4w74hFGe&name=pdf): Dense-MoE and Low-Rank-MoE, which similarly replace all linear layers including those in the attention blocks with an MoE over dense or low-rank matrices. The results show that BTT still has a unique advantage over other structures for constructing structured MoEs, even when applying MoE to all layers. We will include these results in the updated paper.
>
> **On rounding to the nearest integer**. Indeed, rounding to the nearest integers as described in Section 2 can produce a matrix $W$ whose input and output dimensions slightly deviate from the original desired shape. We take the simplest approach to address this issue by padding or truncating the input and output vectors. At scale, the number $\delta N$ of padded or truncated elements becomes vanishingly small relative to the original dimension $N$, as $\delta N / N$ scales as $O(N^{-c})$ where $c$ is the smallest non-zero elements in $\theta$. We will update the paper to clarify this consideration.
>
> Thank you again for your constructive feedback and support. We made a significant effort to address your questions which has improved our work substantially; we would appreciate it if you would consider raising your score in light of our strong response and the significance of our work. We believe this paper will help provide a foundation for a nascent and immensely impactful new research area.
>
> [1] Radford et al. 2019. Language models are unsupervised multitask learners

---

> > ### Comment · Reviewer_Mhte · 2024-08-13
> >
> > Thank you for addressing my concerns. Based on your comments I decided to increase my score.

---

### Author Rebuttal · Authors · 2024-08-07

We thank all the reviewers for their feedback and questions. We provide a general response here and individual replies in separate posts below. Inspired by comments from reviewers, we include multiple new experiments encompassing new datasets, new architectures, and alternative MoE structures that significantly strengthen our findings and demonstrate their applicability in much broader settings.

We appreciate the reviewer's recognition of this work's importance and timeliness. Scaling foundation models is primarily bottlenecked by compute in the dense linear layers, where structured matrices offer promising efficiency gains. Therefore, a comprehensive analysis of structured matrices' potential to enhance dense model scaling laws, along with identifying general properties of structures that correlate with their scaling behavior, is of significant value to the field.

**We highlight several particularly novel and impactful contributions in our work:**

1. We provide a unifying framework to conveniently parameterize a large and continuous space of hardware-efficient structured matrices via Einsums. We show this space contains many popular structured matrices explored in prior works, such as low-rank, Kronecker, Tensor-Train, Monarch, and Block Tensor-Train, while most structures within this space are novel. We further develop an informative taxonomy of the space of Einsums based on key properties relevant to machine learning, including the extent of parameter-sharing, matrix rank, and computational complexity.

2. We perform the first extensive comparison on the compute-optimal scaling laws of structured matrices for language modeling. State-of-the-art large language models (LLMs), including the recent Llama 3.1 405B, are purposely trained to be compute-optimal [1, 2], whereas prior works on training with structured matrices do not compare performance under compute optimality (e.g. by training for too many epochs) [3,4]. Our results therefore provide the more appropriate comparison for evaluating structured matrices in realistic LLM training. Indeed, our results reveal that structures such as Monarch, which significantly outperform dense in other contexts, at best match dense performance under compute-optimal scaling laws.

3. We show that differences in the compute-optimal scaling laws across a wide range of structures are mostly governed by a small number of variables defined in our taxonomy. Small $\omega$ (less parameter sharing) and large $\psi$ (closer to full-rank) reliably led to better scaling laws, while $\nu$ (how dense-like a structure is) can be varied while leaving the scaling laws almost unchanged. These insights will make future search for performant structured matrices significantly more efficient, providing a guiding foundation for this important emerging research area.

4. Guided by the insight that full-rank structures that maximize parameters per unit of compute perform the best, we propose BTT-MoE, a novel Mixture-of-Experts (MoE) architecture obtained by sparsifying computation in the BTT structure, proving to be 440% and 28% more compute-efficient than dense and standard MoE for training GPT-2, respectively.

**We now summarize the new experiments we run inspired by the reviewers’ feedback.**  We present results and figures in the [attached pdf](https://openreview.net/attachment?id=cH4w74hFGe&name=pdf).
1. We verify that our findings regarding the relative performance of different structures continue to hold in the following additional setups:
    - A more standard GPT-2 training setup. We train models up to the size of original GPT-2 Small [5], adopting the original vocabulary with 50,257 tokens and using a context length of 512. The results agree with our findings with in Section 4 which used a reduced vocabulary of 96 tokens and a shorter context length of 128 to save cost.

    - Vision transformers trained with cross-entropy loss for autoregressive image generation on CIFAR-5M.

    - MLP trained with Mean-Squared-Error loss on synthetic data generated by a large and randomly initialized MLP.
We chose CIFAR-5M image generation and the synthetic regression as additional tasks because they contain enough training examples required to study compute-optimal scaling laws, unlike other commonly used datasets such as CIFAR-10 or ImageNet classification.

2. We show that BTT-MoE outperforms two additional MoE architectures: Dense-MoE and Low-Rank-MoE, which similarly replace all linear layers including those in the attention blocks with an MoE over dense or low-rank matrices. The results show BTT has a unique advantage over other structures when used in the proposed structured MoE architecture.

We hope the reviewers can consider these results and clarifications, and the broader context of this work and its significance, in their final assessment.

[1] Dubey et al. 2024. The Llama 3 Herd of Models.

[2] Hoffmann et al. 2022. Training compute-optimal large language models.

[3] Dao et al. 2022. Monarch: Expressive structured matrices for efficient and accurate training

[4] Qiu et al. 2024. Compute Better Spent: Replacing Dense Layers with Structured Matrices

[5] Radford et al. 2019. Language models are unsupervised multitask learners

---

### Decision · Program_Chairs · 2024-09-25

**Decision:**

Accept (poster)

**Comment:**

Authors propose Einstein summation as a framework to generate structured matrices instead of dense layers in transformers architectures and study the loss/ compute scaling laws. Authors identify three parameters of the structured matrix: rank, compute intensity, and parameters per flops that control the scaling laws of the resulting architecture. Authors identify a set of structured matrices  that achieve equal  on par loss as dense layers as a function of training compute. Authors combine these structured matrices with sparse mixture of experts to further improve the scaling law.

Reviewers asked for additional experiments that considered number of factors N>2 and with larger vocabulary and context, authors provided additional results in the rebuttal that satisifed the reviewers with the exception of Reviewer bZBq who still have doubts on the the claim that solely two parameters  parameter sharing and closeness to full-rank reliably led to better scaling laws.

We encourage the authors to revise the paper and include the additional results and to provide more analysis on the effects of the parameters that control the einstein summation.